# Mapping the Featural and Holistic Face Processing of Bad and Good Face Recognizers

**DOI:** 10.3390/bs11050075

**Published:** 2021-05-13

**Authors:** Tessa Marzi, Giorgio Gronchi, Maria Teresa Turano, Fabio Giovannelli, Fiorenza Giganti, Mohamed Rebai, Maria Pia Viggiano

**Affiliations:** 1Section of Psychology—Department of Neuroscience, Psychology, Drug Research and Child’s Health (NEUROFARBA), University of Florence, 12 Via di San Salvi, 50135 Firenze, Italy; tessa.marzi@unifi.it (T.M.); giorgio.gronchi@unifi.it (G.G.); fabio.giovannelli@unifi.it (F.G.); fiorenza.giganti@unifi.it (F.G.); 2Fondazione Turano Onlus, 00195 Roma, Italy; mariateresa.turano@gmail.com; 3Department of Psychology, University of Rouen, 76130 Mont-Saint-Aignan, France; mohamed.rebai@univ-rouen.fr

**Keywords:** individual differences, event-related potentials, featural processing, holistic processing, N170, neural adaptation

## Abstract

Individual abilities in face recognition (good versus bad recognizers) were explored by means of event-related potentials (ERPs). The adaptation response profile of the N170 component to whole faces, eyes and mouths was used in order to highlight the crucial role of individual abilities in identity repetition processes for unfamiliar faces. The main point of this study is to underline the importance of characterizing the performance (bad or good) of the participants and to show that behaviorally selected groups might reveal neural differences. Good recognizers showed selective right hemisphere N170 repetition effects for whole faces and not for features. On the contrary, bad recognizers showed a general repetition effect not specifically related to faces and more pronounced processing for features. These findings suggest a different contribution of holistic and featural analysis in bad and good performers. In conclusion, we propose that the N170 might be used as a tool to tease apart face encoding processes as a function of individual differences.

## 1. Introduction

Faces convey peculiar and biologically relevant information, providing a huge variety of socially relevant signals [1,2,3,4,5,6]. Given the fundamental information conveyed by faces (i.e., age, gender, identity, and mood), face recognition is essential for human social functioning throughout the entire lifespan.

Why do some people recognize faces easily and others frequently make mistakes in recognizing them? People tend to have areas of relative strength and weakness in particular cognitive abilities, and one of the most variable is face recognition. There are huge differences among individuals in the processing of faces, ranging from prosopagnosia (inability to recognize faces) to outstanding individuals with super performances [7,8,9]. In this regard, recent studies have provided evidence for how strongly behavior and brain might be connected by showing that faster face-specific neural processes for structural encoding of faces were associated with better performance in both perceiving and memorizing faces [10]. Different performances have different neural underpinnings and interindividual differences in face recognition are characterized by specific neural signatures [11]. These individual differences form an important phenomenon that has to be considered in face recognition research at behavioral and neural levels.

Face processing has often been associated with specific holistic visual analysis [12,13,14,15]. During holistic processing, facial features (such as eyes, nose, and mouths) are perceived as a combination through which the “whole” face emerges, as a “gestalt”, from the discrete features [16].

The investigation of holistic and non-holistic (featural) face processing is crucial for developing theoretical face recognition models [16], but an exact understanding of facial feature processing involved in the coding of individual faces remains to be established.

An important question is whether featural and holistic processes might be related to interindividual differences in face recognition abilities. Individuals might use different strategies to perform face processing based on their specific face recognition ability. However, although recent studies have begun to uncover evidence that suggests individual differences in face recognition ability are not accompanied by other superior/expert visual skills [17,18], it is still unclear if these variations are specifically related to differences in performing face processing holistically rather than by feature. Results related to the use of these two strategies [19] suggest that all individuals may process faces differently. These differences have been specifically related to face memory performance [19].

In this context, facial feature processing appears essential for subsequent face recognition. Facial features commonly include the nose, mouth and eyes [20], in other words, the parts that correspond to points of discontinuity in the surface geometry of a face [21]. The eyes play a significant role in face perception and can constitute the main reference point for the encoding and retrieval of a new face [22]. Analysis of the eye region is related to individual differences in face recognition abilities. Sekiguchi [23] showed that good face recognizers orient their fixations toward the eyes more than poor face recognizers do. Poor face recognition ability (i.e., acquired prosopagnosia [AP]) has also been related to specific processing of the mouth region [24,25,26]. DeGutis et al. [27] indicated that individuals with congenital prosopagnosia (CP) successfully integrate the mouth but not the eyes into a whole percept. Congenital prosopagnosics also showed reduced holistic face processing, measured with the composite task [28,29,30], indicating that poor face recognition abilities may be associated with compromised holistic processing.

Regarding neurotypical individual differences, Wang et al. [31] reported that face recognition ability increases as the propensity of individuals to process faces holistically increases, measured with both part-whole and composite face tasks. This evidence strongly suggests that facial features and whole face processing shows sensitivity to individual differences in face recognition ability. However, to the best of our knowledge, no studies have yet reported contributions to explore the neural correlates of holistic-feature processing as a function of individual performing characteristics.

Important evidence can be obtained by using event-related potentials (ERPs) that measure an important early face-sensitive component, the N170. The N170 (an incremented negativity 130–200 ms after stimulus presentation over occipito-temporal sites, [32]) appears to be related to the beginning of face-categorization processing as well as the creation of the input for the successive information processing stages [33,34,35]. It is now generally acknowledged that the N170 is not associated with one specific face perception stage but instead reflects the activity of different aspects of face processing, such as detection and subsequent structural encoding [36,37,38]. It has been proposed that the N170 reflects the initial perceptual stage, during which facial features are perceptually “bound” together in a holistic percept [34,39]. This component shows a featural sensitivity: for example, it is larger for isolated eye regions but delayed and smaller for the nose and mouth than for the whole face [32,40,41]. Given the sensitivity of the N170 to both facial features and whole faces, it appears the best candidate for mapping face processing and tapping the neural modulations related to face recognition ability.

We reached the conclusion that it is undoubtedly important to better understand how face recognition is shaped by individual face perceptual abilities, aware that performance in face tests can be extremely variable.

It is crucial to explore whether bad and good face recognizers engage different neural mechanisms to perform face processing [42]. The adaptation paradigm in an ERP study is a suitable procedure to uncover the nature and dynamics of the face recognition process.

Neural adaptation (or repetition suppression) reflects the reduction of neural activity in response to target stimuli when these are preceded by physically or categorically identical adaptor stimuli.

It was shown that rapid successive presentation of two stimuli attenuates the N170 (and M170) response. Such effect is face-selective, with greater attenuation when faces are preceded by other faces rather than by objects [43,44,45]. These paradigms have been previously employed with success in fMRI studies investigating the neural responses of face-specific brain regions [46] and electrophysiological studies examining the time course of face processing [43,47,48]. Some ERP studies have quantified the N170 adaptation by looking at the N170 modulations triggered by the second stimulus presented [38] and interpreted the amplitude reductions as a detection mechanism with decreasing neural responses’ redundancy and increasing coding efficiency [47].

Specifically, the present study explored whether whole faces and distinct facial features might elicit equivalent N170 adaptation effects in bad and good face recognizers. We compared whole faces with face components such as eyes and mouths. Eyes appear to be a key diagnostic feature for recognizing gender, identity, and facial expressions [49], and a high face expertise appears to be related to an increased ability to extract information from the eye area [50]. By contrast, poor face recognizers seem to process the mouth more [24,25,51].

We classified bad and good face recognizers based on a specifically designed selection process that relied on both self-report scoring (IFA-Q) [11,48] and the Cambridge Face Memory Test (CFMT) [7]. We hypothesized a possible differential sensitivity to facial features, with an advantage in good face performers for whole faces.

In the present study, we used such a combined approach with the twofold aim (i) of achieving greater understanding in both the neural and functional mechanisms of face cognition and (ii) highlighting the underlying sources of individual variation in face recognition abilities.

## 2. Material and Methods

### 2.1. Participants

Ninety students (70 women, mean age = 26.62, SD = 7.2) of the University of Florence participated in the study. They were given the Italian Face Ability Questionnaire (IFA-Q) and the CFMT.

Twenty-seven participants were then selected to participate in the EEG-ERPs experiment based on their high (self-evaluated as a good face recognizer) and low (self-evaluated as a bad face recognizer) scores on the IFA-Q [11]. Objective face recognition ability was measured with the CFMT [7]. Based on their z-CFMT score, participants were divided into two groups: bad recognizers and good recognizers. The former scored below one SD from the mean (*n* = 15, six women, 12 right-handed, mean age = 25.2, SD = 5). The average total score out of 72 for bad face recognizers was 49.3 (SD = 4). Good face recognizers scored above one SD from the mean (*n* = 12, eight women, 11 right-handed, mean age = 24, SD = 4.31). The average total score out of 72 for good face recognizers was 67.4 (SD = 2.5). This integrated methodological approach [11,48] permitted us to obtain an adequate sample size (13% of the total sample distribution) to identify reliable differences.

None of the participants had a history of neurological, ophthalmological, or psychiatric disease and all had normal or corrected-to-normal vision. Additionally, none of them reported taking medication. They signed written informed consent and received class credit for their participation. The study complied with the Declaration of Helsinki. All data were collected and processed anonymously. This study is part of a set of behavioral and non-invasive studies on face recognition processing approved by the Research Committee of the University of Florence (protocol number 17245_2012).

#### 2.1.1. Participant Selection: IFA-Q and CFMT

##### The Italian Face Ability Questionnaire (IFA-Q)

The IFA-Q [11,48] comprises 25 forced-choice items on a 4-point Likert scale (1: I strongly agree; 2: I agree; 3: I disagree; and 4: I strongly disagree). First, participants were required to answer open questions about the presence of face recognition disorders in their families, severe vision issues, head trauma, and the presence of neurological or psychiatric problems. The items are based on the life experience of typical face performers and the symptoms usually associated with CP [28,52,53,54,55,56,57,58]. Additionally, specific items are aimed at estimating the use of coping or compensatory strategies. The majority of the items dealt with the evaluation of specific face patterns, such as facial expressions processing, part-based processing, and eye gaze contact. The range varied from 25 to 100: low scores indicated potentially poor face recognizers and high scores good face recognizers. IFA-Q is characterized by good reliability (Cronbach’s alpha = 0.84) and validity (correlation between IFA-Q score and the Cambridge Face Memory Test, Spearman rho = 0.5, *p* < 0.001).

##### Cambridge Face Memory Test (CFMT)

In the CFMT participants are asked to recognize six learned faces in three phases: the same images (introduction), different perspectives of the same images, and different perspectives of the same images covered with heavy visual noise. Initially, participants are required to memorize the six target faces and then are tested with forced-choice items comprising a target and two distracters. For each target face, three test items contain views identical to those studied in the introduction, five present new views, and four present new views with noise (for details, see [7]).

Overall, the criteria employed for group differentiation were based on a preliminary, self-reported measure aimed to detect in a large sample good and bad faced recognizers and then on an objective-task validation of the face-recognition performance of the two groups. This method has been already adopted and validated [11,48] being able to efficiently detect differences between good and bad face recognizers.

### 2.2. Stimuli

The stimuli images (half female, half male) consisted of 120 whole faces, 120 eyes, and 120 mouths created using FaceGen Modeller 3.2; a total of 360 distinct identities were generated. They were presented at fixation, with eye gaze straight ahead (for eye and whole face stimuli) and no facial expression of emotions, against a dark background. Whole faces had a size of 238 × 280 pixels, sustaining a visual angle of 6.35° × 7.50°. The dimensions of eyes and mouths were 204 × 72 pixels, sustaining a visual angle of 5.44° × 1.95°. All images were converted to greyscale with the same contrast and luminance values via Adobe Photoshop. All stimuli were comparable in brightness and aligned on a dark equiluminant background. We employed E-prime software (Psychology Software Tools, Pittsburgh, PA, USA) for stimulus presentation and behavioral response recording.

### 2.3. Task

The EEG recording was performed in a comfortable room while the participant was sitting in front of a computer monitor (dimensions: width 340 mm × height 275 mm; resolution: width 1280 pixels × height 1024 pixels) at a distance of 57 cm and performed a same/different matching task. After filling in the informed consent form, the electrode cap for EEG recording was set and participants were given task instructions and practice trials. They were asked to minimize blinking and to keep their eyes on a small fixation cross in the center of the screen during the EEG recording.

In each trial, two stimuli (two whole faces, two eyes, or two mouths) were presented in rapid succession (S1–S2), separated by an inter-stimulus interval of 200 ms: participants had to perform a same/different matching task on the second stimulus. After a fixation cross (lasting 500 ms), the adaptor stimuli (S1) appeared for 500 ms; the target-test stimuli (S2) were presented for 200 ms (see Figure 1). Participants were asked to press a response key labelled “same” for trials where the stimuli pair showed the same individual face or features and “different” for stimuli from different individuals. A training block containing two pairs of stimuli (one same and one different) was delivered before the first experimental block. In order to avoid repetitions, all stimuli were displayed only once. The study included three experimental blocks with 120 trials each (20 same and 20 different for each stimulus category), with a pause at the end of each block. In total, we presented 60 same and 60 different trials for each stimulus category. All trials and blocks were presented in random order. ERP times were time-locked to the repetition of the same pair of stimuli.

### 2.4. EEG Data Recording

An EEG was continuously recorded via 28 Ag/AgCl electrodes (F7, F3, FZ, F4, F8, FT7, FC3, FCZ, FC4, FT8, T3, C3, CZ, C4, T4, TP7, CP3, CPZ, CP4, TP8, T5, P3, PZ, P4, T6, O1, Oz, O2) with NeuroScan 4.3 and amplified using the NuAmp system. Electrooculograms (vertical and horizontal) were recorded bipolarly from the outer canthi of both eyes. Two electrodes, one placed on the left mastoid and one on the right, served as references for online recording. EEGs were re-referenced offline to the average of the left and right mastoids. For all electrodes, the impedance was maintained at less than 5 KΩ.

### 2.5. EEG Data Analysis

Electrical activity was amplified by using a bandpass of 0.01–100 Hz and a sampling rate of 1000 Hz. EEGs were epoched offline from 100 ms before to 500 ms after stimulus onset. All epochs with ocular artifacts larger than 40 μV were rejected; the remaining epochs were averaged separately for each experimental condition and low-pass filtered at 25 Hz with a zero-phase shift digital filter and re-referenced offline. The averages were computed for trials where the eye, mouth, or whole face pairs were behaviorally classified correctly as “same.” All ERPs were computed relative to a 100 ms pre-stimulus baseline. After artifact rejections, the trials included in the average counts were limited to correct trials. A similar number of correct trials was observed across the conditions, ranging from 45 to 51 trials. ERP components were identified based on visual inspection and the literature regarding face-related ERP components.

## 3. Results

### 3.1. Behavioral Results

Bad and good face recognizers presented a high task performance and similar RTs. The mean accuracy for the good recognizers was 94% (sd = 5.9) for the whole face category, 93% (sd = 6.3) for the eye, and 89% (sd = 7.1) for the mouth. For the bad recognizers, it was 94% (sd = 6.0) for the whole face category, 92% (sd = 5.9) for the eye, and 92% (sd = 8.0) for the mouth. The mean RTs for the good recognizers were 376 ms (sd = 78.3) for the whole face category, 378 ms (sd = 90.3) for the eye, and 384 ms (sd = 107.1) for the mouth. For the bad recognizers, 379 ms (sd = 100.4) for the whole face category, 389 ms (sd = 91.1) for the eye, and 370 ms (sd = 98.2) for the mouth. We compared the percentages of correct responses and mean reaction times (RTs) by means of ANOVA. No significant effect was found (all ps > 0.1), probably as a consequence of a ceiling effect.

### 3.2. Electrophysiological Results: Responses to S2 Stimuli

The time window of the N170 was selected based on the topographical distribution and centered on their maximum amplitude. The N170 was analyzed for the temporal electrodes (T5 and T6). The mean amplitude in the time window between 150 and 260 ms was calculated and used as the dependent variable. The adaptation effect was evaluated by using a repeated-measures ANOVA considering the correct responses to the S2, with Group (Bad, Good) as a between-factor, Adaptation (S1, S2), Category (Whole Face, Eye, Mouth) and Hemisphere (T5 and T6) as within-factors analysis.

Given that Mauchly’s test of Sphericity indicated a violation of sphericity assumption (Chi_(2)_^2^ = 8.98, *p* = 0.011), in the following analyses, the Greenhouse–Geisser correction was adopted and adjusted degrees of freedom were used. The Bonferroni correction was performed in the case of multiple comparisons.

In the overall analysis of the N170 amplitude, there were significant main effects of Adaptation and Category (F(2, 25) = 12.08, *p* = 0.002, η_p_^2^ = 0.32., F(2, 50) = 4.48, *p* = 0.003, η_p_^2^ = 0.15).

These effects were further qualified by the significant interaction Adaptation × Category × Hemisphere × Group (F(2, 50) = 7.08, *p* = 0.001, η_p_^2^ = 0.24). Post hoc analyses were performed with additional ANOVAs. Figure 2 shows the grand averages for the stimuli, divided for good (top panel) and bad recognizers (bottom panel). A general effect of adaptation that is very sensitive to individual perceptual abilities is shown in Figure 3. When the N170 decreases in its amplitude in S2 compared to S1, then it is possible to assume possible effects of adaptation or repetition. This happens in good recognizers specifically for whole faces, whereas for bad recognizers there is an overall repetition effect for all categories.

Bad performers showed reduced amplitude to S2 compared to S1 which was not related to a specific category, indexing a general repetition effect (F(1, 14) = 5.62, *p* = 0.03, η_p_^2^ = 0.28). A different pattern of results emerged for good performers, showing reduced amplitudes for S2 whole faces compared to S1whole faces (F(2, 22) = 12.73, *p* = 0.001, η_p_^2^ = 0.54). Moreover, S2 whole faces were reduced in amplitude compared to eyes (*p* < 0.01) and compared to mouths (*p* < 0.001). This difference between S1 and S2 for whole faces was more pronounced in the right than in the left hemisphere (F(2, 22) = 5.55, *p* = 0.001, η_p_^2^ = 0.34). Figure 4 depicts the mean amplitude values of the N170 for the single subjects, the 12 good performers on the top and the 15 bad performers on the bottom.

Furthermore, in the right hemisphere a decreased amplitude was found for S2 whole faces compared to S1 whole faces (F(2, 50) = 19.65, *p* = 0.0001, η_p_^2^ = 0.44). This effect was further characterized by the comparison between groups: the decrease in amplitude for S2 whole faces was greater in good compared to bad recognizers (F(2, 50) = 9.4, *p* = 0.0001, η_p_^2^ = 0.27). In addition, bad recognizers showed a diminished amplitude to mouths (S2) compared to good recognizers (*p* < 0.01). In the left hemisphere, S1 stimuli showed enhanced amplitudes compared to S2 stimuli (F(1, 25) = 6.93, *p* = 0.001, η_p_^2^ = 0.22).

No differences between categories were found for S1, whereas S2 whole faces were reduced compared to whole faces S1, reflecting possible repetition or adaptation effects (F(1, 25) = 27.52, *p* = 0.0001, η_p_^2^ = 0.52, see Figure 3). In S2, mouths were more enhanced in good compared to bad recognizers (F(2, 50) = 7.61, *p* = 0.003, η_p_^2^ = 0.21).

No significant interactions with the factor Group were obtained for the P100 and N250 components (although some effects are visible in Figure 2).

Topographical maps showing scalp distribution of the N170 for the different categories and in the two groups are reported in Figure 5.

The figure shows scalp topography of the N170 component in response to stimuli presented at S1 and S2, separately for the good and bad performers groups, based on amplitudes obtained in the measurement interval of the N170 (150 and 260 ms). Scalp distribution of the N170 is similar for both groups, but N170 amplitudes are clearly attenuated for whole faces in good recognizers in S2 (repetition effect). Another notable effect is the enhanced amplitude for mouth in good compared to bad recognizers in the right hemisphere in S2.

## 4. Discussion

In this study, we aimed to characterize, by means of ERPs, the sensitivity of the N170 component to whole faces (indexing holistic processing) or to eyes and mouths (indexing feature-based processing) depending on different recognition capacities. Specifically, we explored the response profile of the N170 component in participants who were previously selected (by using the CFMT and the IFA-Q) for their bad or good face recognition performance.

Investigation of both holistic and featural face processing has recently been referred to as crucial for developing theoretical face recognition models [16]. To further pursue the study of this specific issue, we considered individual differences as a key to better understanding the nature of early face processes. This study was motivated by the hypothesis that the holistic or featural sensitivity of the N170 component might be substantially different in good and bad recognizers. We interpreted differences in the N170 adaptation or repetition effect as evidence of N170 sensitivity to a specific facial category (i.e., whole face, eyes, or mouth).

The adaptation effect of repeated stimuli in rapid succession generally involves the engagement of the same neural population for both adaptor (S1) and target stimuli (S2) [43,45,47,49,59]. In this study, we used the repetition effect of the N170 as a possible tool to tap differences related to two different groups of participants that differed in their ability to recognize unfamiliar faces. These results extend our previous study [11] by highlighting possible dissociable visual neural strategies (holistic or featural) used by face recognizers.

At a behavioral level, bad and good face recognizers presented a high performance and similar RTs, probably as a consequence of a ceiling effect. Another possible interpretation is to consider these results as reflecting the everyday life behaviors of bad and good face recognizers. Differently from congenital prosopagnosics [60], bad face recognizers do not present particular difficulties related to their scarce face-recognizing ability, therefore behavioral differences did not show up in task performance.

Face recognition tasks have been used behaviorally to explore the role of individual differences in holistic face processing [31,61,62,63], but it is still an open issue whether individuals who are bad or good at face recognition effectively engage different neural mechanisms to perform face processing [42].

The N170 in response to S1 stimuli partially fits with the classic view that N170 reflects the initial perceptual face encoding [34,39]. The N170 showed a sensitivity for eyes and whole faces as adaptors [32,34,39,64,65], as well as for the mouth category. According to the results of Harris and Nakayama [43] that showed an adaptation effect of the M170 (the magnetoencephalographic N170 counterpart) to upright faces that was robust when they and facial features (eyes) were used as adaptors; our results extend this finding to the mouth category. Harris and Nakayama [43] also suggested that the M170 is exclusively sensitive to facial features rather than facial configuration (i.e., whole face), and it is therefore likely to be generated at stages that precede face-specific holistic processing. However, the present results suggest that N170 might also reflect processes that are sensitive to the configuration of whole face stimuli.

A distinctive pattern of the N170 amplitude related to individual differences was observed for S2 stimuli. In good face recognizers, in the right hemisphere, the N170 adaptation effect for whole faces was greater relative to that of eyes and mouths. Good recognizers showed a face specificity at the level of the N170 in the right hemisphere. On the contrary, bad recognizers showed an overall repetition suppression-/adaptation- effect but not specific for whole faces. Therefore, it is possible that the structural encoding processes are more sensitive to specific holistic face processing in good face recognizers than in bad. We hypothesized that good recognizers are more attuned to holistic processing in the right hemisphere, where featural and holistic information are encoded together in a unique face representation [39,62,64,66,67,68]. In bad face recognizers, there were no differences in the N170 adaptation effect, suggesting the activation of neural patterns that respond similarly to whole faces and facial features. Good face recognizers probably take strength in coding the gestalt of the face at an initial perceptual stage, as proposed by Yovel [69], which emphasizes that a holistic representation of a face is generated at 170 ms after stimulus onset over the right hemisphere. In agreement, recent results revealed that the N170 component does not exclusively reflect the detection of individual face parts, but analysis of configural and global aspects of faces as well [70]. It might be the case that good performers are more efficient in adopting a holistic processing strategy based on the integration of internal features [14].

Our results suggest that interindividual abilities in face recognition could be rooted in the early stages of face processes, which might start around 170 ms, involving different strategies during encoding.

At an earlier level, considering the P100 component, we did not find remarkable differences, indexing a general mechanism of detection of features and configurable aspects. This idea is supported by the spatio-temporal dynamics reported by Negrini and colleagues [71] in which the P100 is sensitive to both facial features and spacing between features.

Considering the response patterns of bad recognizers, we can speculate that their abilities recruit the right hemisphere also for facial features because they are less sensitive to configural whole faces. This interpretation is strengthened by the result that showed, in the right hemisphere in bad recognizers, a possible repetition effect for mouths (and not whole faces). This result is reminiscent of behavioral patterns found in individuals with congenital/developmental prosopagnosia (CP). An impairment in face processing, shown on the M170, to early perceptual stages in holistic processing was found in CP [72]. Another aspect to consider is that in the face part-whole effect [13], which underlies a better performance in neurotypical individuals in matching facial features when these are displayed enclosed in a face rather than when they are shown in isolation, individuals with CP have a better performance in the matching of mouths. No advantages of the face context were observed in CP individuals when matching the eye region [27]. Bad face recognizers could present a similar pattern with a perceptual system more attuned to the mouth region compared to good face recognizers. Similar to prosopagnosics, bad face recognizers could be less able to integrate features in a complex face pattern to reach a global percept. Specifically, they could lack the ability to integrate information about local features with information about global shapes. When eye and mouth features are presented together in a whole face, these features could compete with a right-hemisphere process more focused on the mouth.

We interpret our results in line with the view that the N170 reflects the activity of different aspects of face processing [34,36,37,38,39,73]. Overall, based on our results, we suggest that the N170 is a manifold component linked to certain distinct aspects of face processing. Indeed, at a generic level, “the N170 might be associated with the activation of face-selective neurons triggered whenever a face or part of a face is present in the visual field and reflects a relatively early stage in the structural encoding of face parts” (p. 2451) [34]. Moreover, at a more detailed level (N170 when stimuli are repeated), the pattern of the N170 adaptation effects observed in this study provides evidence that holistic and featural sensitivity might be associated with individual differences.

It is worthy to note that also other ERPs face-sensitive components have emerged in the face literature [74]. After the N170, at occipito-temporal, the P200 has been suggested to be sensitive to face typicality and to second-order spatial relations [45]. Going on in the processing, the N250 has been related to face repetitions and familiarity representing a tool that discriminates familiar from unfamiliar faces [75]. Finally, semantic processing has been related to the N450 [73] and to the late positive potential [45].

A limitation of our experimental design was that it was not suitable for exploring the effects on the N250r component. We think that at this processing level, very important insights might emerge to better clarify the activation of memory representations of individual faces [76].

Previous studies showed the importance of facial features in driving the early neuronal response to faces [77,78,79]. In this context, we would like to highlight the importance of holistic or feature analysis as a prerogative of individual differences: bad and good face recognizers might share the same process of face detection at the N170 level [41,65] but might differ when a more detailed visual processing has to be carried out, as in this case, with a matching task after the stimuli repetition.

Further studies are necessary to characterize more precisely these results by exploring the N250r (as well as other face-sensitive components) and using non-face stimuli as controls. A study of the time course of face recognition in individuals with different abilities might yield information about the nature of face recognition.

## 5. Conclusions

We examined whether featural and holistic information makes distinct contributions to face processing based on interindividual variability. Our study suggests that the early neural coding of a face, as indexed by the N170, is modulated by individual differences in face recognition abilities. The present results suggest consideration of face abilities at both a behavioral and neural level when dealing with processes involved in face perception. Individuals with good face recognition abilities are better attuned and more specific in representing faces holistically. In contrast, those who are bad at recognizing faces are probably not well specialized for whole faces and are more attracted to analyzing features.

Unfamiliar face recognition is generally challenging and requires much effort given the enormous number of everyday faces we encounter; in particular, bad recognizers are certainly disadvantaged in this domain.

Our results can give useful insights within the brain–computer interface literature and, more generally, the application of machine learning to EEG signals data. Since we observed that the same N170 component may reflect radically different kinds of processing depending on good and bad face recognizers, future technologies should begin to take into account also individual differences. Indeed, exploiting other relevant factors (individual variations, other EEG signals, and so on) could help to attribute more precise meaning to the same input. In addition, these findings highlight the crucial role of interindividual face differences in understanding how the human brain orchestrates the operations that characterize face recognition.

## Figures and Tables

**Figure 1 behavsci-11-00075-f001:**
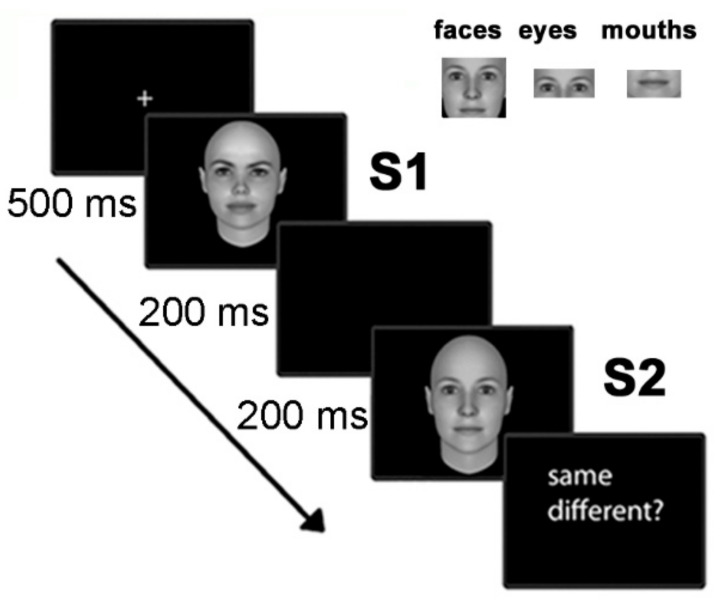
The procedure: A same/different matching task for repeated stimuli S1 and S2; the stimuli consisted of whole faces, mouths or eyes.

**Figure 2 behavsci-11-00075-f002:**
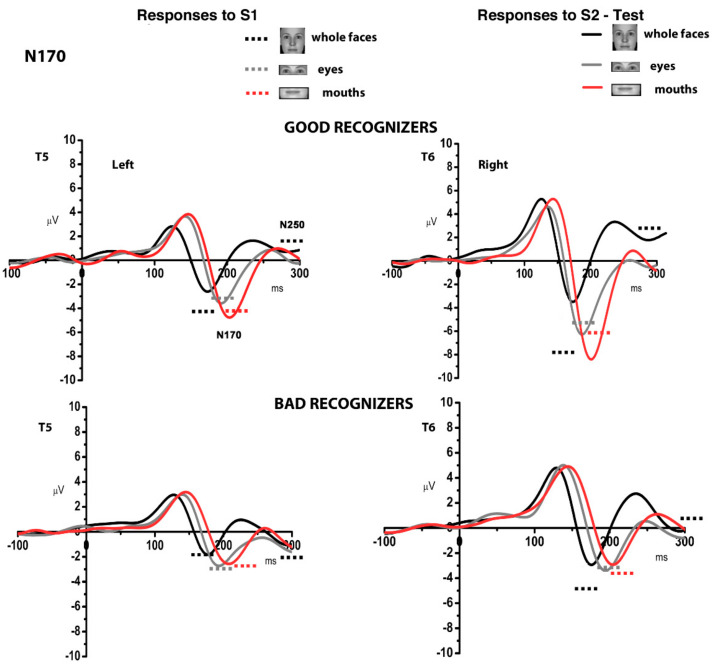
Grand average responses (T5 and T6 electrodes) for good and bad face recognizers for the different categories. The continuous lines represent responses to S2 while the dots represent the mean amplitudes of the N170 and (the N250) for responses to S1.

**Figure 3 behavsci-11-00075-f003:**
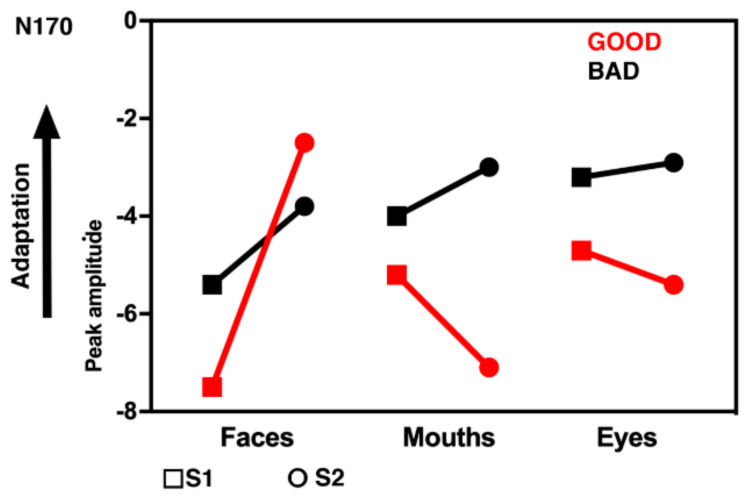
The adaptation effect of the N170 for bad and good recognizers: S1 is confronted with S2 for faces, mouths and eyes.

**Figure 4 behavsci-11-00075-f004:**
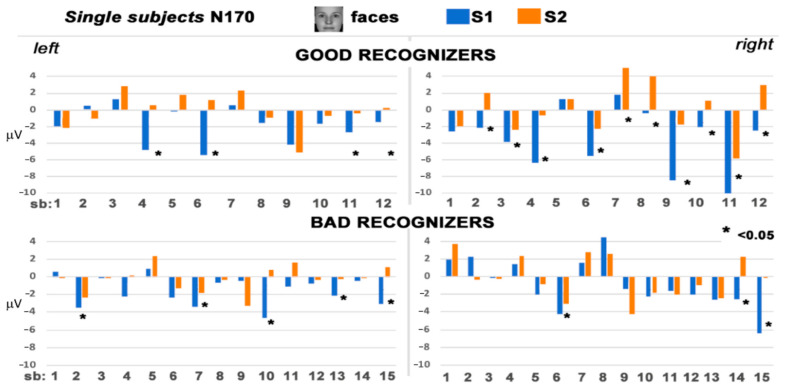
Single subjects (top panel good recognizers, bottom panel bad recognizers) mean amplitude values for whole faces in S1 and S2, measured in the time range of the N170 component.

**Figure 5 behavsci-11-00075-f005:**
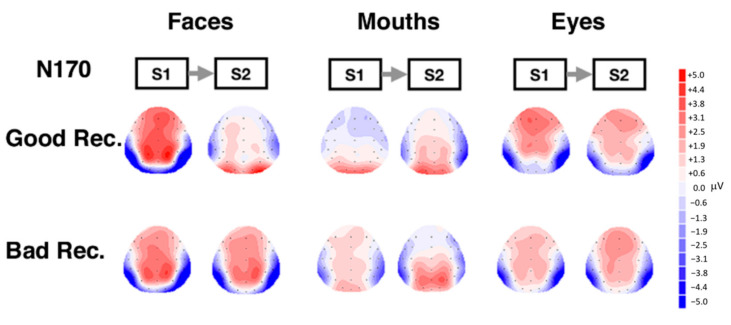
Topographical maps showing scalp distribution for the N170 component. Maps were generated by event-related potentials (ERPs) mean amplitudes measured in the N170 time window for S1 and S2.

## Data Availability

Data are available from the corresponding author on request.

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
