# Peer review of "Mapping the Featural and Holistic Face Processing of Bad and Good Face Recognizers"

_behavsci, 2021, doi:10.3390/bs11050075_

Round 1

Reviewer 1 Report

Individual differences in face processing performance must correspond with difference in neural processing, but these neural differences are not well understood. The authors present novel EEG research to test for hypothesized differences in the 170ms that would indicate differences in early stimulus encoding processes. This hypothesis is motivated with respect to known individual differences with respect to feature-based vs. holistic face processing, with more capable face-processors typically perceiving faces more holistically.

The paper is clear and the experiment is well motivated is clearly execute and reported. There are some minor points that could be more clear.

  1. I am unclear about the stimuli. There were 120 trials (40 faces, 40 eyes, 40 mouths) in each of three experiment blocks, and 360 "identities". What are "identities"? Are the eyes and mouths taken from the 120 faces, or are 360 distinct faces generated, with 120 cropped to just the eyes and another 120 cropped to just the mouth?
  2. The variance of the behavioral results should be reported (despite insignificant statistical tests). For that matter, the statistical procedure should be described (even if it's very simple).
  3. The dependent variable for the ERP ANOVA is ambiguous. Is it the mean amplitude in the 150 -- 260 ms window?
  4. The ANOVA focuses specifically on electrodes T5 and T6, but figures seem to report grand averages (by hemisphere). Is there a reason for the discrepancy?

The results are thoughtfully and conservatively discussed.

Author Response

Response to Reviewer 1 Comments

Individual differences in face processing performance must correspond with difference in neural processing, but these neural differences are not well understood. The authors present novel EEG research to test for hypothesized differences in the 170ms that would indicate differences in early stimulus encoding processes. This hypothesis is motivated with respect to known individual differences with respect to feature-based vs. holistic face processing, with more capable face-processors typically perceiving faces more holistically.

The paper is clear and the experiment is well motivated is clearly execute and reported. There are some minor points that could be more clear.

  1. I am unclear about the stimuli. There were 120 trials (40 faces, 40 eyes, 40 mouths) in each of three experiment blocks, and 360 "identities". What are "identities"? Are the eyes and mouths taken from the 120 faces, or are 360 distinct faces generated, with 120 cropped to just the eyes and another 120 cropped to just the mouth?

Response: With the term “identities” we mean distinct identities, with 120 stimuli showing the whole face, 120 cropped to just the eyes and 120 cropped to just the mouth. We clarified this point in the stimuli section.

  1. The variance of the behavioral results should be reported (despite insignificant statistical tests). For that matter, the statistical procedure should be described (even if it's very simple).

Response: We added the variance of the behavioral results and the statistical procedure.

  1. The dependent variable for the ERP ANOVA is ambiguous. Is it the mean amplitude in the 150 -- 260 ms window?

Response: The dependent variable for the ERP (N170) ANOVA was the mean amplitude in the time window between 150 and 260 ms. This detail is now clearly reported.

  1. The ANOVA focuses specifically on electrodes T5 and T6, but figures seem to report grand averages (by hemisphere). Is there a reason for the discrepancy?

Response: As reported in the text 3.2 section the ANOVA focuses specifically on electrodes T5 and T6. This information is now added in the Figure 2 legend.

The results are thoughtfully and conservatively discussed.

Reviewer 2 Report

Mapping the featural and holistic face processing of bad and  good face recognizers

16/04/2021

The present work shows a study that tries to analyze how individual differences could affect facial recognition. The work deals with a topic of great interest.

The authors do a good job of theoretically contextualizing the subject under study.

Aspects for improvement:

  • I believe that the objectives and hypotheses raised in the study could be stated more clearly.
  • There are some typographical errors in the text. The text needs a good editing.
  • The criteria used for the differentiation of the groups based on the scores recorded would need further justification.
  • Evidence should be provided on the validity and reliability of the tests used in the study.
  • It is surprising to me that there are no differences in behavior between good and bad recognizers. Is there an explanation for this lack of effect?
  • Greenhouse – Geisser correction and and adjusted degrees of freedom needs to be justified
  • Figures 4 and 5 should have a legend for the axes

Author Response

Response to Reviewer 2 Comments

The present work shows a study that tries to analyze how individual differences could affect facial recognition. The work deals with a topic of great interest.

The authors do a good job of theoretically contextualizing the subject under study.

Aspects for improvement:

I believe that the objectives and hypotheses raised in the study could be stated more clearly.

Response: We described the objectives and hypotheses better at the end of the introduction.

There are some typographical errors in the text. The text needs a good editing.

Response: The text was reviewed but a native English speaker

The criteria used for the differentiation of the groups based on the scores recorded would need further justification.

Response: we added a clarification about the differentiation of the groups.

Evidence should be provided on the validity and reliability of the tests used in the study.

Response: we added more information about the validity and the reliability of the IFA-Q.

It is surprising to me that there are no differences in behavior between good and bad recognizers. Is there an explanation for this lack of effect?

Response: We attributed the lack of a difference to a ceiling effect related to the ease of the task. Indeed, the accuracy was very high (about 90% for both conditions) and RTs relatively fast. Noteworthy, in everyday life bad and good face recognizers do not have a sensible difference. As we wrote in the discussion, differently from congenital prosopagnosics, bad face recognizers do not present particular difficulties related to their scarce face ability, therefore, behavioral differences may not emerge in behavioral task.

Greenhouse – Geisser correction and and adjusted degrees of freedom needs to be justified

Response: We justified the adoption of the Greenhouse – Geisser correction and the  adjusted degrees of freedom

Figures 4 and 5 should have a legend for the axes

Response: We added the legend for the y-axes of Figures 4 and 5.

Reviewer 3 Report

This manuscript address a very trendy topic ounderlying Good and bad recognizers in terms of the N170.

I congratulate the authors as the paper is well-structured and conducted.

I only have minor points up to the authors, that I hope can add value to the current findings:

-It would be interesting to decribe other waves related to face recognition in the introduction, as well as for future lines of reseach (future replications of your work), as described in previous literature:

"Face processing is evidenced temporally through ERP Components, described as the N170 for structural encoding (Bentin, Allison, Puce, Perez, & McCarthy, 1996), the N250 for familiar faces, and N450 for semantic face processing (Rossion & Jacques, 2011)." from Moret-Tatay, C., Fortea, I. B., & Sevilla, M. D. G. (2020). Challenges and insights for the visual system: Are face and word recognition two sides of the same coin?. Journal of Neurolinguistics56, 100941.

-It would be interesting to enphasize the applied level of your results.

Author Response

Response to Reviewer 3 Comments

This manuscript address a very trendy topic ounderlying Good and bad recognizers in terms of the N170.

I congratulate the authors as the paper is well-structured and conducted.

I only have minor points up to the authors, that I hope can add value to the current findings:

-It would be interesting to decribe other waves related to face recognition in the introduction, as well as for future lines of reseach (future replications of your work), as described in previous literature:

"Face processing is evidenced temporally through ERP Components, described as the N170 for structural encoding (Bentin, Allison, Puce, Perez, & McCarthy, 1996), the N250 for familiar faces, and N450 for semantic face processing (Rossion & Jacques, 2011)." from Moret-Tatay, C., Fortea, I. B., & Sevilla, M. D. G. (2020). Challenges and insights for the visual system: Are face and word recognition two sides of the same coin?. Journal of Neurolinguistics, 56, 100941.

Response: According to the Referee’s suggestion, we added in the Discussion the following sentence: It is worthy to note that also other ERPs face-sensitive component have emerged in the face literature [74]. After the N170, at occipitotemporal the P200 has been suggested to be sensitive to face typicality and to second-order spatial relations [45]. Going on in the processing, the N250 has been related to face repetitions and familiarity representing a tool that discriminates familiar from unfamiliar faces [75]. Finally, semantic processing has been related to the N450 [73] and to the late positive potential [45].

-It would be interesting to enphasize the applied level of your results.

Response: We added a section describing the applied level of our results in the conclusion.
